# Gut Microbial Composition and Liver Metabolite Changes Induced by Ammonia Stress in Juveniles of an Invasive Freshwater Turtle

**DOI:** 10.3390/biology11091315

**Published:** 2022-09-05

**Authors:** Han Li, Qinyuan Meng, Wanling Wang, Dongmei Mo, Wei Dang, Hongliang Lu

**Affiliations:** Key Laboratory of Hangzhou City for Ecosystem Protection and Restoration, School of Life and Environmental Sciences, Hangzhou Normal University, Hangzhou 311121, China

**Keywords:** *Trachemys scripta elegans*, ammonium exposure, growth rate, gut microbiota, liver metabolome

## Abstract

**Simple Summary:**

The ammonia content in aquaculture waterbodies can easily achieve a relatively high level, largely due to the excretion of intensively farmed aquatic animals and the decomposition of residual food particles. Excess ammonia may exert prominent toxicities to all aquatic species. Many freshwater turtles have become economically important aquaculture species in some countries. The evaluation of the toxic effects of ammonia on these cultured turtle species is rarely considered, despite having been extensively conducted in many invertebrates and fish. Various omics techniques can be applied to comprehensively analyze diverse biological responses at different levels in pollutant-exposed individuals. In order to increase our understanding of the physiological changes under ammonia stress in cultured turtle species, we exposed juveniles of the red-eared slider turtle (*Trachemys scripta elegans*) to different concentrations (0, 0.3, 3.0, and 20.0 mg/L) of ammonium chloride solution for 30 days, and then measured their swimming performance, growth rate, gut microbial composition, and liver metabolic profiling. No significant change in swimming ability was found in ammonia-exposed turtles, but a relatively low growth rate was exhibited in the 20.0 mg/L ammonia-exposed turtles. Although no significant differences in the alpha and beta diversity measures of the gut microbial community were observed, the microbial composition seemed to be different among different treatment groups. Some pathogenic bacteria were found to increase in ammonia-exposed turtles. Liver metabolic profiling based on liquid chromatography–mass spectrometry showed that a series of metabolites (such as leucine, valine, arginine, glutamine, adenosine diphosphate, glyceric acid, myo-inositol, gamma-aminobutyric acid, etc.) were significantly altered in ammonia-exposed turtles. Accordingly, ammonia exposure might alter the health status of guts due to increased pathogenic bacteria, and disturb multiple metabolic pathways (such as amino acid, nucleotide, lipid metabolism, energy metabolism, etc.) in juveniles of *T. scripta elegans*.

**Abstract:**

As the most common pollutant in aquaculture systems, the toxic effects of ammonia have been extensively explored in cultured fish, molluscs, and crustaceans, but have rarely been considered in turtle species. In this study, juveniles of the invasive turtle, *Trachemys scripta elegans*, were exposed to different ammonia levels (0, 0.3, 3.0, and 20.0 mg/L) for 30 days to evaluate the physiological, gut microbiomic, and liver metabolomic responses to ammonia in this turtle species. Except for a relatively low growth rate of turtles exposed to the highest concentration, ammonia exposure had no significant impact on the locomotor ability and gut microbial diversity of turtles. However, the composition of the microbial community could be altered, with some pathogenic bacteria being increased in ammonia-exposed turtles, which might indicate the change in their health status. Furthermore, hepatic metabolite profiles via liquid chromatography–mass spectrometry revealed extensive metabolic perturbations, despite being primarily involved in amino acid biosynthesis and metabolism. Overall, our results show that ammonia exposure causes gut dysbacteriosis and disturbs various metabolic pathways in aquatic turtle species. Considering discrepant defense mechanisms, the toxic impacts of ammonia at environmentally relevant concentrations on physiological performance might be less pronounced in turtles compared with fish and other invertebrates.

## 1. Introduction

Due to the decomposition of nitrogenous organic compounds and the excretion of aquatic organisms, ammonia nitrogen (ammonia-N) can normally occur in natural waterbodies (<1.5 mg/L in the surface water of China’s major rivers, according to China’s national surface water quality report). However, the content of ammonia-N in aquaculture wastewater easily achieves a relatively high level. For example, ammonia-N levels in some shrimp-cultured ponds can reach as high as 20 mg/L [1]. Excess ammonia-N in waterbodies would cause serious toxicity to aquatic organisms. In cultured shrimps, high environmental ammonia exposure has been shown to induce cell apoptosis, alter immune-related gene expression, lead to histological damage, hinder growth, and even cause death [2,3,4,5,6,7].

The toxicological effects of ammonia exposure have been well explored in cultured fish and crustacean species [7,8,9,10,11,12,13,14,15,16]. As important aquaculture species, some freshwater turtles have been widely cultured in some countries. However, studies considering the impacts of aquaculture environmental pollutants, including ammonia-N, remain limited to only a few turtle species [10,17,18,19,20,21]. Specifically, the physiological responses and defense mechanisms to ammonia stress are relatively better understood only in the aquatic Chinese soft-shelled turtle, *Pelodiscus sinensis* [17,22,23,24,25], but not in other semi-aquatic turtle species. The results of recent studies show that the physiological and biochemical responses to ammonia stress can vary considerably among different semi-aquatic turtle species [17]. For that reason, exploring diverse biological responses at different levels should still be a necessary task when improving our understanding of physiological defense mechanisms under environmental-pollutant stresses in turtle species.

With the rapid development of various omics techniques, comprehensively analyzing diverse responses at different levels in aquatic organisms under exposure to relatively low doses of environmental pollutants has become simpler and easier to achieve [26]. These techniques have been applied only in a few studies conducted on aquatic cultured species [27]. In this study, we measured the gut microbiome and liver metabolome responses of juvenile red-eared slider turtles (*Trachemys scripta elegans*) under exposure to different concentrations of ammonia, in order to evaluate ecotoxicological effects on this semi-aquatic turtle species. As one of the most successful invasive species worldwide, it could be expected that *T. scripta elegans* might have a great resistance to ammonia stress relative to other species. Nevertheless, a previous study showed that exposure to high concentrations of ammonia would cause marked toxic effects on various tissues in *T. scripta elegans*, such as damaging intestinal structure and altering its microbial community [18]. Investigating other physiological responses, such as metabolome responses in various tissues, may contribute to unraveling more mechanisms behind the toxic actions of ammonia, and the physiological adaptations to ammonia in this species. We expected that some intestinal bacteria and liver metabolites could be altered to a certain extent in ammonia-exposed *T. scripta elegans* juveniles.

## 2. Materials and Methods

### 2.1. Experimental Animals and Ammonia Exposure Treatment

Fertilized eggs of *T. scripta elegans* were purchased from a private farm in Huzhou city, Zhejiang province in June 2021, and incubated in 25 × 20 × 10 cm^3^ plastic containers with moist vermiculite. Containers were placed in a temperature incubator (Binder KB240, Binder GmbH, Tuttlingen, Germany) set at 28 °C, and distilled water was added to the vermiculite if necessary. Newly hatched turtles were then collected and reared in glass tanks.

Approximately three months after hatching, a total of 20 juvenile turtles were randomly selected, weighed, and equally allocated to different treatments (control-CTRL, 0.3, 3.0, and 20.0 mg/L ammonia, 5 individuals in each treatment). The concentration of 3.0 mg/L ammonia selected in this study is the maximum dose of the primary standard of requirement for water discharge from freshwater aquaculture ponds (SC/T 9101-2007), whilst the concentration of 20.0 mg/L is the reported maximum level in some cultured ponds [1]. Turtles were housed individually in aquaria, which contained equal amounts of dechlorinated tap water. Experimental treatment concentrations were prepared and adjusted by adding ammonium chloride (NH_4_Cl, purity ≥99.5%, CAS No. 12125-02-9) stock solutions. During the exposure experiment, turtles were fed a commercially available diet, daily. Full water changes were performed every other day.

### 2.2. Swimming and Growth Performance

Swimming performance of each turtle was measured on the day before ending the exposure. Turtles were introduced individually into a bath (120 × 10 × 20 cm^3^, placed in a climate-controlled room at 24 °C) with a 10 cm depth of water (its temperature was maintained at the corresponding level), they were encouraged to swim, and then their swimming performance was recorded with a Panasonic HDC-SD900GK digital video camera. The video clips were examined later for the average speed over 50 cm.

After 30 days of exposure, the turtles were reweighed, euthanized on ice, and dissected. Their guts and livers were isolated and transferred to centrifuge tubes. After being weighed, these tissues were frozen in liquid nitrogen and then stored at −80 °C. We calculated the specific growth rate of the individuals during exposure using the following formula: specific growth rate = (ln*W*_t_ − ln*W*_0_)/T × 100%, where *W*_0_ = initial body mass, *W*_t_ = final body mass [28].

### 2.3. Gut Microbiota

Gut bacterial DNA was extracted from the collected intestinal contents using the Qiagen TM QIAamp DNA Stool Mini Kit (Qiagen GmbH, Hilden, North Rhine Westphalia, Germany). The V3–V4 variable region of the 16 S rRNA gene was amplified using the primers B341F (5′- CCTACGGGNGGCWGCAG -3′) and B785R (5′- GACTACHVGGGTATCTAATCC -3′). Two rounds of PCR were performed here. The first round of PCR mix (25 µL) contained 1.25 µL DNA templates (10 ng/µL), 0.25 µL of each primer (25 µM), 10.75 µL grade water, and 12.5 µL 2 × KAPA HiFi HotStart Ready Mix (KAPA Biosystems, Wilmington, MA, USA), and the thermal cycling conditions were: 95 °C for 3 min, followed by 25 cycles at 95 °C for 30 s, 55 °C for 30 s, 72 °C for 30 s, and the final extension at 72 °C for 5 min. The second round of PCR mix (25 µL) contained 2.5 µL of the purified products from the first round of PCR, 0.25 µL of each primer, 9.5 µL grade water, and 12.5 µL 2 × KAPA HiFi HotStart Ready Mix, and the thermal cycling conditions were: 95 °C for 3 min, followed by 8 cycles at 95 °C for 30 s, 55 °C for 30 s, 72 °C for 30 s, and the final extension at 72°C for 5 min. Purified amplicons were sequenced on the Illumina NovaSeq PE250 platform (Illumina Inc., San Diego, CA, USA) by the Hangzhou Kaitai Biotechnology Co., Ltd. (Hangzhou, China).

### 2.4. Hepatic Metabolic Profiling

Approximately 100 mg of the liver tissue was weighed from each sample and then transferred into a 2 mL centrifuge tube, where pre-chilled methanol–chloroform mixture, double-distilled water, and steel balls were added. After being ground up, ultra-sonicated and incubated on ice, the samples were centrifuged (12,000 rpm) for 10 min at 4 °C. Then, supernatants were transferred into clean centrifuge tubes and dried. Their residue was finally dissolved with pre-chilled 2-chlorobenzalanine/50% acetonitrile solution and filtrated through a 0.22 μm filtering membrane prior to liquid chromatography–mass spectrometry (LC-MS) analysis.

Chromatographic separation was accomplished with a Thermo Ultimate 3000 platform equipped with an Acquity UPLC^®^ HSS T3 (2.1 × 150 mm, 1.8 μm) column, and mass spectrometry analysis was carried out on a Q Exactive mass spectrometer (Thermo, San Jose, CA, USA). The temperature of the separation column was set at 8 °C, and the flow rate at 250 μL/min. An amount of 2 μL from each sample was then added into the separation column after equilibration. The mobile phase comprised a mixture of 0.1% formic acid in water and 0.1% formic acid in acetonitrile (the positive ion mode), and 5 mM ammonium formate in water and acetonitrile (the negative ion mode). The elution program was: 2% formic acid in acetonitrile from 0 to 1 min, increased to 50% formic acid in acetonitrile from 1 to 9 min, increased to 98% formic acid in acetonitrile from 9 to 12 min and maintained at this level until 13.5 min, decreased to 2% formic acid in acetonitrile from 13.5 to 14 min and maintained at this level until 17 min for the positive ion mode; 2% acetonitrile from 0 to 1 min, increased to 50% acetonitrile from 1 to 9 min, increased to 98% acetonitrile from 9 to 12 min and maintained at this level until 13.5 min, decreased to 2% acetonitrile from 13.5 to 14 min and maintained at this level until 17 min for the negative ion mode. The parameters of mass spectrometer were: spray voltage, 3.5 kV (positive mode) and −2.5 kV (negative mode); sheath gas, 30 arbitrary units; auxiliary gas, 10 arbitrary units; and capillary temperature, 325 °C. The analyzer scanned over a mass range of *m/z* 100–1000 at 60,000 resolution. The full-scan mass spectral data were acquired through a high-energy collisional dissociation (HCD) scan with a normalized collision energy of 30 eV. Some unnecessary information was removed using dynamic exclusion. LC-MS analysis was performed at Suzhou PANOMIX Biomedical Tech Co., Ltd. (Suzhou, China).

### 2.5. Data Processing and Analysis

One-way analysis of variance (ANOVA) was performed to determine the differences in initial body mass, swimming speed, and growth rate of juvenile turtles.

Raw paired-end reads of gut microbiota were quality filtered and processed with cutadapt and VSEARCH. Effective sequences were clustered into operational taxonomic units (OTUs) at the 97% similarity level. Representative sequences were annotated by searching the RDP and SILVA databases, and the α diversity (Chao and Shannon–Wiener) indexes were calculated for each sample using the Mothur software. Non-parametric Kruskal–Wallis tests were performed to test the among-group differences in α diversity indexes and gut microbial composition at each taxonomic level, and principal coordinates analysis (PCoA) based on unifrac distances was performed to test among-group differences in the relative abundance of OTUs for the β diversity analysis.

Hepatic metabolomic data processing was performed as previously described [29]. Multivariate unsupervised principal component analysis (PCA) and supervised partial least squares–discriminant analysis (PLS-DA) was performed to test among-group differences in hepatic metabolites. Metabolites were identified by searching the mass spectra against available databases (i.e., Human Metabolome Database (HMDB), Metabolite Link (Metlin), MassBank Database). One-way ANOVAs were performed to determine among-group differences in key identified metabolites.

## 3. Results

### 3.1. Swimming Speed and Growth Rate

Prior to exposure, the body mass of juvenile turtles did not differ among treatment groups (*F*_3,16_ = 0.99, *p* = 0.421). Although 20 mg/L ammonia-exposed turtles seemed to gain mass slowly compared with other groups during exposure, no statistically significant differences could be found among treatment groups in swimming speed (*F*_3,16_ = 0.42, *p* = 0.744) and growth rate (*F*_3,16_ = 2.47, *p* = 0.099) (Figure 1).

### 3.2. Gut Microbiota Composition

Twenty intestinal content samples were used for the bacterial composition analysis. Each sample contained at least 89,900 effective sequences. The rarefaction curves based on the Chao and Shannon–Wiener indexes indicated that these sequence depths were sufficient in each sample (Appendix A. Rarefaction curves of Chao and Shannon–Wiener indexes for all samples). However, no significant among-group differences in the α diversity of gut microbiota could be found here (Kruskal–Wallis test, Chao index: *H*_3, *N* = 20_ = 2.25, *p* = 0.523, Figure 2A; Shannon–Wiener index, *H*_3, *N* = 20_ = 2.20, *p* = 0.532, Figure 2B). Similarly, no significant separations of bacterial communities among groups were identified by the unweighted (Figure 2C) or weighted (Figure 2D) PCoA of β diversity.

Further taxonomic analysis showed that Firmicutes (61.0 ± 6.5%), Bacteroidetes (22.8 ± 3.9%), and Proteobacteria (12.8 ± 5.5%) were the most predominant phyla (Figure 3A). Although the relative abundance of Firmicutes seemed to have decreased, while that of Proteobacteria had increased with increasing ammonia concentration, among-group differences were not statistically significant in these phyla (Kruskal–Wallis test, all *p* > 0.353). At the family level, the relative abundance of some bacterial families were different among groups (Figure 3B). For example, the relative abundance of Enterobacteriaceae appeared to be higher in the 20 mg/L-exposed group (22.32 ± 15.00%) than in other groups (CTRL: 0.06 ± 0.04%; 0.3 mg/L-exposed: 0.71 ± 0.32%; 3.0 mg/L-exposed: 0.47 ± 0.24%) (Figure 3B); although some pathogenic bacterial families (such as Aeromonadaceae and Pseudomonadaceae) accounted for a very low proportion, they were mainly found in high-concentration exposure groups (e.g., 0.06 ± 0.03% and 0.04 ± 0.02% in the 20 mg/L-exposed group). At the genus level, more than 30 genera were identified. Of them, *Clostridium_sensu_stricto* (15.50 ± 3.84%), *Bacteroides* (10.29 ± 3.31%), *Romboutsia* (9.06 ± 4.26%), *Terrisporobacter* (2.62 ± 0.87%), *Parabacteroides* (1.69 ± 0.70%), *Hungatella* (1.53 ± 0.69%), *Turicibacter* (1.51 ± 0.55%), *Monoglobus* (1.21 ± 0.52%), and some unclassified genera belonging to Lachnospiraceae (14.24 ± 3.45%), Bacteroidales (8.94 ± 3.40%), Enterobacteriaceae (5.81 ± 4.06%), Clostridiales (3.06 ± 0.88%), and Clostridiaceae_1 (1.41 ± 0.38%) were the most predominant genera (Figure 3C,D). The relative abundance of some identified bacterial genera differed among the treatment groups. For example, despite accounting for a very low proportion, the relative abundances of some pathogenic bacterial genera, such as *Aeromonas* (CTRL vs. 20 mg/L-exposed group: 0.002 ± 0.002% vs. 0.06 ± 0.02%) and *Pseudomonas* (0 vs. 0.02 ± 0.01%), were increased in high-concentration-exposed turtles (Figure 3D).

### 3.3. Hepatic Metabolite Profile

The PCA showed a degree of separation among different treatment groups, with the first two components explaining 34.6% and 37.6% of the total variance of the raw data in the positive and negative ion modes, respectively (Figure 4A,B). The PLS-DA further identified these differences and revealed a more obvious among-group separation (Figure 4C,D).

Compared with the CTRL group, 61, 125, and 166 metabolites changed remarkably in the 0.3, 3.0, and 20 mg/L ammonia-exposed groups, respectively (Figure 5). Some differentially expressed metabolites at different ammonia concentrations were identified, and these metabolites were primarily involved in amino acid metabolism (such as leucine, valine, tryptophan, serine, citrulline, arginine, glutamine, glutamate, etc.). These amino acids decreased somewhat with increasing ammonia concentration (Figure 6). Similarly, hepatic adenosine diphosphate (ADP), uridine, glyceric acid, myo-inositol, and γ-aminobutyric acid levels also decreased when exposed to high levels of ammonia concentration (Figure 6).

## 4. Discussion

Toxic effects induced by ammonia exposure have been documented in various aquatic organisms [14,19,30]. Under exposure to higher concentrations of ammonia, animal growth was significantly reduced, which was observed in some fish species [31,32,33]. Although the mean growth rate of 20 mg/L ammonia-exposed turtles was relatively low (a decrease of approximately 74% compared with the CTRL ones, Figure 1), it did not differ significantly from that of the CTRL ones. No striking effect of ammonia exposure on growth rate was also found in another turtle species, *P. sinensis* [22]. Similarly, decreased locomotor ability caused by ammonia exposure has been highlighted in fish species [34], but not in *T. scripta elegans*. Considering the above differences, a relatively higher tolerance ability to ammonia could be expected in turtle species.

Decreased intestinal microbial community diversity in ammonia-exposed *T. scripta elegans* was shown in a previous study [18], but was not found in the present study. Discrepant findings might partially result from the differences in ammonia exposure level, the age stage of experimental animals, and the rearing environmental condition in the two studies (e.g., a relatively higher ammonia level: approximately 32 mg/L total ammonia-N, and older individuals were used in the previous study [18]). In fact, between-species comparison also displayed inconsistent ammonia impacts on intestinal microbial diversity. For example, exposure to different levels of ammonia decreased the microbial community diversity of Pacific white shrimps, *Litopenaeus vannamei* (20 mg/L total ammonia-N, [5]), Asian clams, *Corbicula fluminea* (25 mg/L, [35]), yellow catfish, *Pelteobagrus fulvidraco* (146 mg/L, [36]), and striped-neck turtles, *Mauremys sinensis* (200 mg/L, [20]), but had no significant impact on crucian carps, *Carassius auratus* (50 mg/L, [37]). Accordingly, the toxic effects of ammonia exposure on gut microbiota could not be simply reflected by reduced microbial diversity.

Furthermore, the bacterial community composition was found to change significantly after ammonia exposure. As reported in other turtles [20,28], Firmicutes was the most dominant phylum in the guts of *T. scripta elegans*. Despite having no statistical difference, the relative abundance of Firmicutes appeared to decrease with increasing ammonia levels. Firmicutes is considered to be involved in nutrient uptake [38]. The abundance of intestinal Firmicutes might affect the growth rate of the host, and that was partially reflected by the lowered growth rate of 20 mg/L ammonia-exposed *T. scripta elegans*. Contrarily, the relative abundance of Proteobacteria appeared to increase in ammonia-exposed turtles. Proteobacteria includes a wide variety of pathogens, and its prevalence may potentially increase the risk of disease in aquatic organisms [38,39]. Specifically, some pathogenic bacterial genera belonging to Proteobacteria, such as *Aeromonas* and *Pseudomonas*, were significantly increased in ammonia-exposed turtles, potentially indicating an unhealthy intestinal state. Other metabolism- or immunity-related bacterial genera, such as *Romboutsia* and *Clostridium_sensu_stricto*, were shown to decrease in ammonia-exposed *T. scripta elegans* in the previous study [18]. However, among-group differences in the abundances of these genera were not statistically significant, which might be due to relatively lower ammonia levels used here.

As reported in other aquatic species [7,12,16,40], a series of identified metabolites were shown to change significantly in the livers of ammonia-exposed *T. scripta elegans*. The alterations in these hepatic metabolites appeared to be dose-dependent, and primarily reflected the perturbations in amino acid biosynthesis and metabolism. For example, the decreased level of branch-chain amino acid (BCAA) in livers of ammonia-exposed turtles, despite being not statistically significant in isoleucine, might be the result of accelerated catabolism of BCAAs associated with liver injury [41]. Some amino acids, such as arginine and citrulline, would contribute to partially alleviating environmental ammonia stress through their role in the ornithine–urea cycle [42]. An observed reduction in these amino acid levels reflected the occurrence of inhibitions in amino acid metabolic pathways related to ammonia metabolism in ammonia-exposed *T. scripta elegans*, as well as in fish [36]. Under ammonia stress, increasing the synthesis of glutamine from glutamate and ammonia by the action of glutamine synthetase is another major mechanism of ammonia assimilation, and has been reported in some ammonia-exposed aquatic species [13,16,43]. Contrarily, the synthesis of glutamine seemed to be hindered in ammonia-exposed turtles because of an observed reduction in hepatic glutamine (and glutamate) levels in this study. Glutamate and γ-aminobutyric acid (GABA) are important amino acid neurotransmitters [44]. The alterations of these metabolites are believed to be associated with toxicant-induced neurological disorders, although elevated levels of glutamate (in fish brains) or GABA were observed after pollutant exposure in some aquatic organisms [14,43]. On the other side, GABA also shows a protective effect on ammonia-induced liver damage, and could reduce tissue ammonia content [45,46]. How GABA influences the metabolic pathways related to ammonia metabolism needs to be further investigated. However, we could speculate that the reduced hepatic GABA level here might suggest a deficiency caused by excessive consumption.

Levels of several hepatic metabolites associated with energy metabolism were shown to change after ammonia exposure in some aquatic species, such as crustaceans and fish [7,12,14,16,40,47]. Despite having only a marginal difference, a key tricarboxylic acid (TCA) cycle intermediate, malate, was shown to decrease in the livers of ammonia-exposed turtles, probably indicating that a perturbation related to the functioning of the TCA cycle occurred in this species after ammonia exposure. In fact, ammonia-induced perturbations in the TCA cycle were observed in all of the above-mentioned studies. Some amino acids can participate in energy metabolism through specific metabolic pathways, such as being converted into the TCA cycle intermediates [48]. For example, leucine may enter the TCA cycle by being converted to citrate [7,29]. Accordingly, reductions in relevant amino acids in this study would potentially exert adverse impacts in these metabolic pathways and thus disturb energy metabolism. On the other side, the reduction in some hepatic amino acids might also be due to their excessive consumption as a result of increased energy demand for animals under ammonia stress [12,16]. Ammonia-induced perturbation in energy metabolism was also reflected by the observed alteration in the glycolytic metabolites, such as glyceric acid [47].

Other disorders of metabolism, such as nucleotide and lipid metabolism, may also occur in various tissues of ammonia-exposed aquatic organisms [7,12,16]. The reduction in some catabolites of nucleotide metabolism here (e.g., myo-inositol) might be related to the disorder of nucleotide metabolism. Overall, ammonia exposure might cause comprehensive metabolic disturbances in the liver as well as in other tissues of *T. scripta elegans*. Nevertheless, the exploration of ammonia-induced metabolic changes in various aquatic organisms might be still required to reveal the physiological mechanism of the toxic actions of ammonia in further studies.

## 5. Conclusions

In this study, general physiological performance, intestinal microbial community composition, and hepatic metabolic response were investigated in ammonia-exposed red-eared slider turtles. After ammonia exposure, observable changes in growth rate, intestinal microbial composition, and hepatic metabolites were exhibited in this species, reflecting its potential toxic effects. The health status of ammonia-exposed turtles might be influenced because some pathogenic bacteria were significantly increased in their guts. Hepatic metabolomic profiling showed a series of metabolite changes, indicating that ammonia exposure would exert comprehensive toxic effects in this species. Various ways of improving ammonia assimilation and/or secretion could be adopted by turtle species [10,21]. Compared with fish and aquatic invertebrates, a relatively high tolerance to ammonia might be exhibited in turtle species, which was implied by no striking changes in functional performances in *T. scripta elegans*, as well as in other species (e.g., *P. sinensis*, [22]). These results might improve our understanding of physiological changes under environmental stresses in aquatic animals, especially in invasive turtle species potentially having physiological superiority over native species.

## Figures and Tables

**Figure 1 biology-11-01315-f001:**
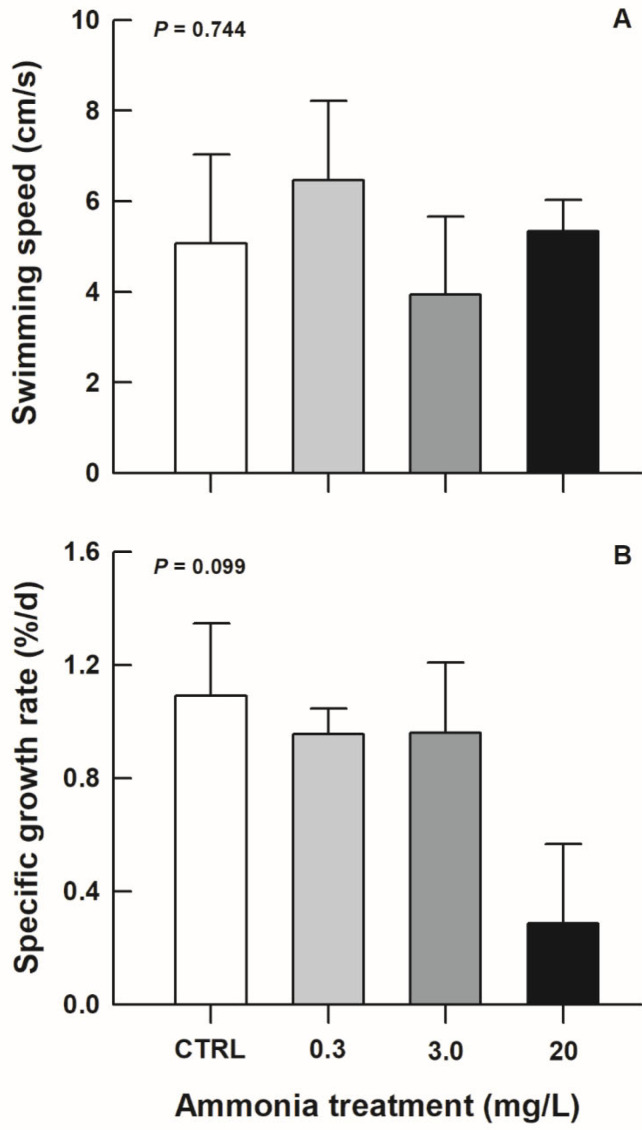
Swimming speed (**A**) and specific growth rate (**B**) of *Trachemys scripta elegans* juveniles exposed to CTRL, 0.3, 3.0, and 20 mg/L ammonia.

**Figure 2 biology-11-01315-f002:**
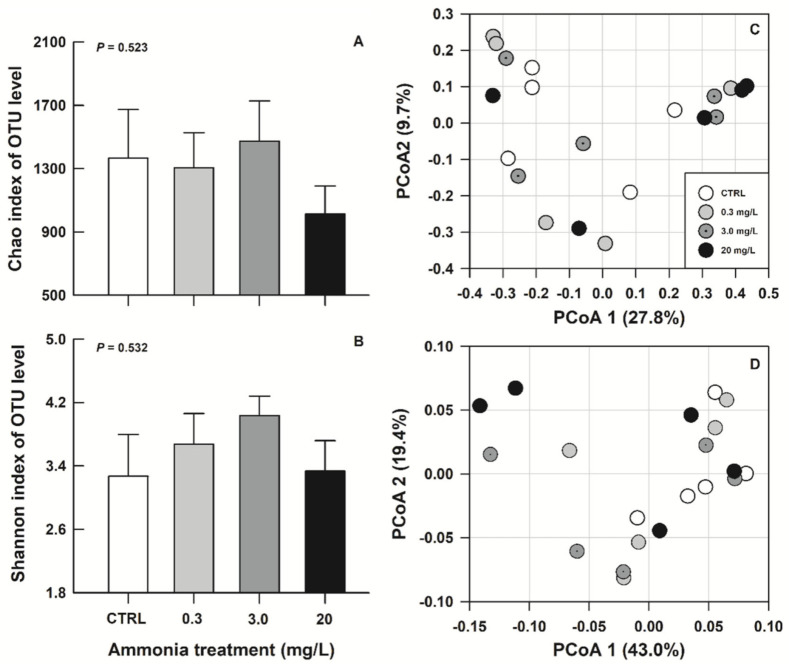
The Chao (**A**), Shannon–Wiener (**B**) index, and score plots for unweighted (**C**) or weighted (**D**) principal coordinates analysis for gut microbiota of *Trachemys scripta elegans* juveniles exposed to CTRL, 0.3, 3.0, and 20 mg/L ammonia.

**Figure 3 biology-11-01315-f003:**
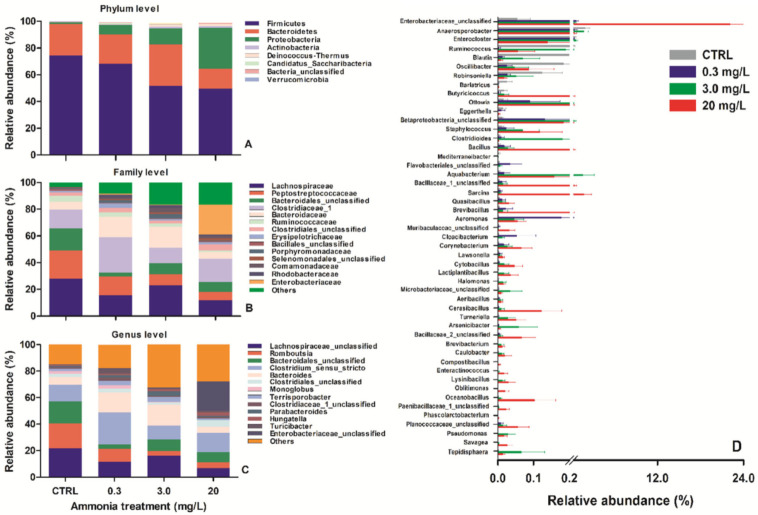
Relative abundance of the gut microbiota at the phylum (**A**), family (**B**), and genus (**C**) levels, and significantly changed bacterial genera (**D**) in different treatment groups of *Trachemys scripta elegans* juveniles.

**Figure 4 biology-11-01315-f004:**
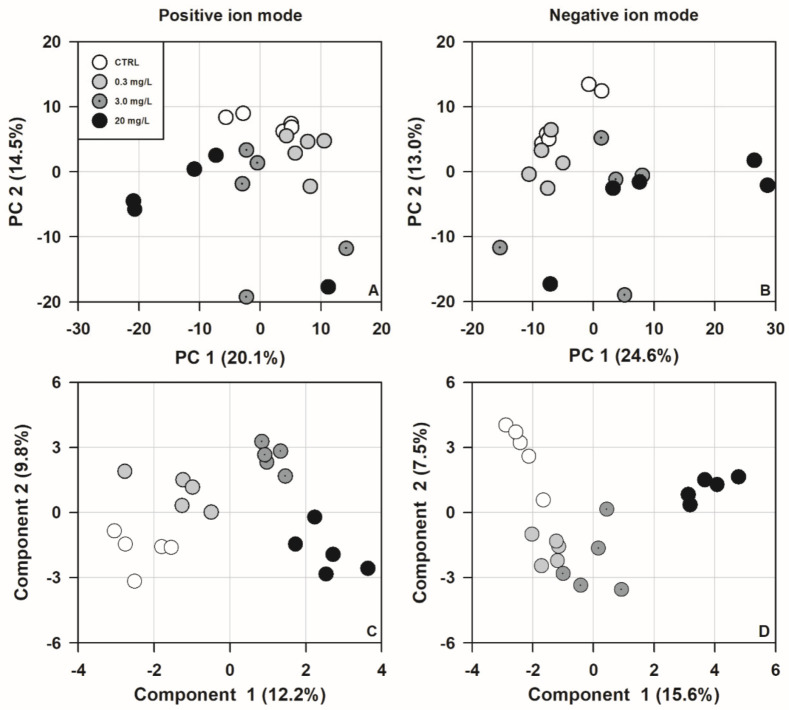
Score plots for principal component analysis (PCA), in the positive (**A**) and negative (**B**) ion mode, and partial least squares–discriminant analysis (PLS-DA), in the positive (**C**) and negative (**D**) ion mode of hepatic metabolite profiles showing separation among different treatment groups of *Trachemys scripta elegans* juveniles.

**Figure 5 biology-11-01315-f005:**
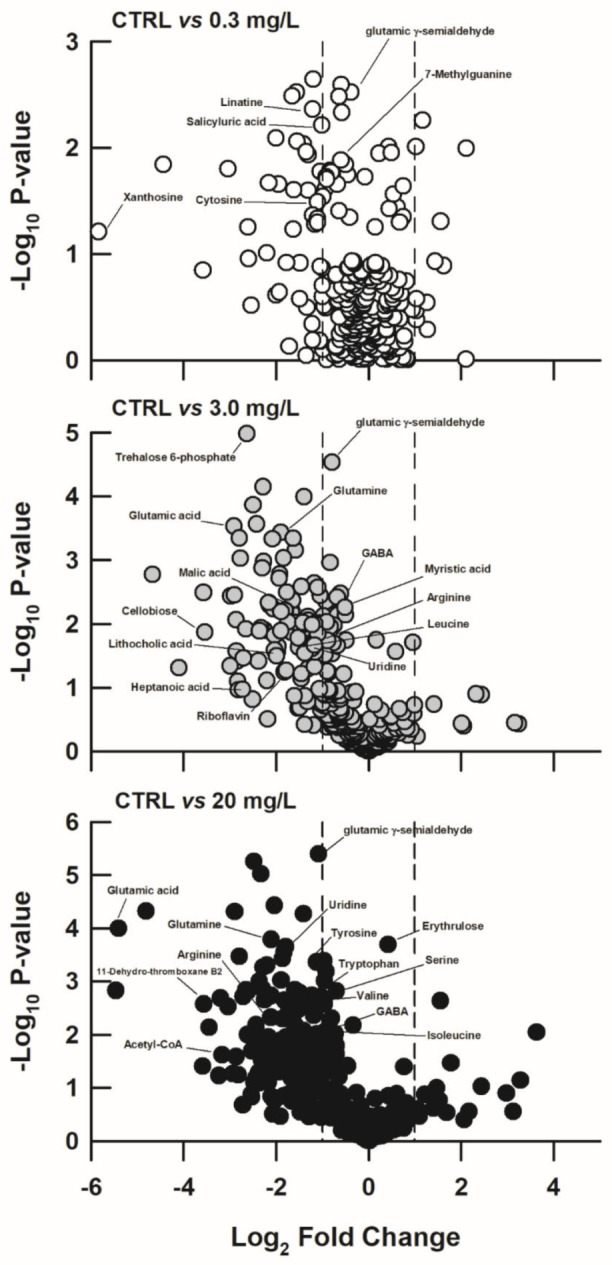
Volcano plots showing several differential hepatic metabolites between the CTRL and 0.3 mg/L, between the CTRL and 3.0 mg/L, and between the CTRL and 20 mg/L ammonia-exposed group of *Trachemys scripta elegans* juveniles.

**Figure 6 biology-11-01315-f006:**
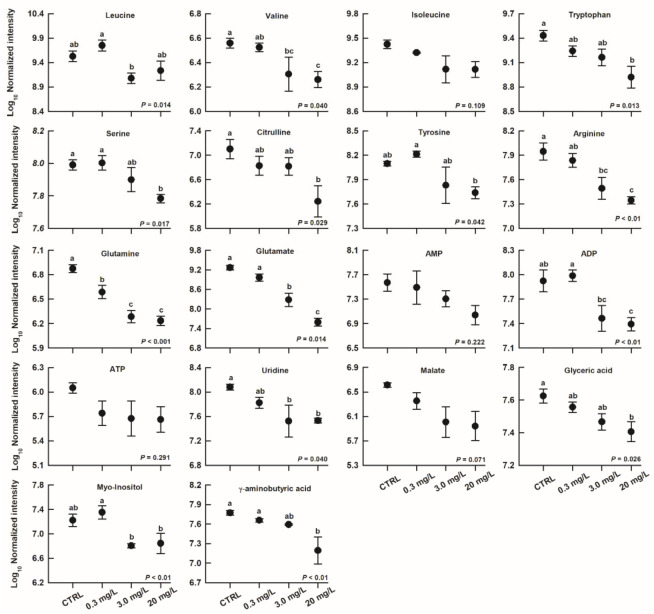
Mean values (±standard error) for several identified hepatic metabolites of *Trachemys scripta elegans* juveniles exposed to CTRL, 0.3, 3.0, and 20 mg/L ammonia. Different letters indicated groups that differed significantly (a > b > c).

## Data Availability

All data generated by this study are available in this manuscript and the accompanying Appendix A.

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
