# Peer review of "Gut Microbial Composition and Liver Metabolite Changes Induced by Ammonia Stress in Juveniles of an Invasive Freshwater Turtle"

_biology, 2022, doi:10.3390/biology11091315_

Round 1

Reviewer 1 Report

The work by Li et al. entitled “Gut Microbial Composition and Liver Metabolite Changes Induced by Ammonia Stress in an Invasive Freshwater Turtle” is a very interesting physiological study, exploring new insights into ammonia toxicity effects in key tissues of vertebrates. The design and the writing is solid, justified and focused. Therefore, I can suggest nothing more than the publication of this manuscript.

For the authors convenience, I suggest the following comments (if they wish to address them):

-Some minor grammatical and syntactical errors could be minimized.

-The manuscript would benefit a lot if the authors added in the end of the manuscript, a graphical (something like a graphical abstract) resuming all their work. This acts like a take-home message to the reader

-Add some future perspectives for this turtle species and the research concerning it in conclusions. Just 1 or 2 short sentences.

Author Response

The work by Li et al. entitled “Gut Microbial Composition and Liver Metabolite Changes Induced by Ammonia Stress in an Invasive Freshwater Turtle” is a very interesting physiological study, exploring new insights into ammonia toxicity effects in key tissues of vertebrates. The design and the writing is solid, justified and focused. Therefore, I can suggest nothing more than the publication of this manuscript.

For the authors convenience, I suggest the following comments (if they wish to address them):

Some minor grammatical and syntactical errors could be minimized.

Response 1: We read through the manuscript, and corrected some grammatical and syntactical errors.

The manuscript would benefit a lot if the authors added in the end of the manuscript, a graphical (something like a graphical abstract) resuming all their work. This acts like a take-home message to the reader

Response 2: Thanks for the reviewer’s kind suggestion. According to the requirements of the journal, Simple Summary was added at the beginning of the revised version of manuscript, which may have a similar effect as a graphical resuming at the end of MS. So, it was not added in the revised version of MS.

Add some future perspectives for this turtle species and the research concerning it in conclusions. Just 1 or 2 short sentences.

Response 3: Thanks. As suggested, relevant sentences were added at the end of the Conclusion.

Reviewer 2 Report

Li et al. evaluated the physiological, gut microbiomic and liver metabolomic responses to chronic exposure of different ammonia levels (0, 0.3, 3.0, and 20.0 mg/L) in juveniles of invasive turtle, Trachemys scripta elegans. They got the conclusion that ammonia exposure would significantly influence gut health, and disturb various metabolic pathways in aquatic turtle species. They measured gut microbiomic responses to chronic ammonia exposure, but did not test intestinal structure, so the conclusion should be specific.

 This group has found that exposure to high concentrations of ammonia would cause marked toxic effects on various tissues including damaging intestinal structure and altering its microbial community in T. scripta elegans. The rationales to study responses to chronic ammonia exposure in juveniles of Trachemys scripta elegans should be mentioned in the introduction.

In this study, the authors just tested the responses to ammonia levels in juveniles of invasive turtle and these results could not reflect the responses of all age stages, so juveniles should be present in the title.

The statistical analysis was missed in the part of Material and Methods. The authors should describe the statistical analysis as an independent part in details.

 What’s the statistical method used by the authors for analyzing the data of microbial composition? There are diverse bacteria in the gut and they can influence each other. Did they use adjusted P values?

 It’s the similar condition for the metabolites. It’s better to correct the statistical analysis.

 For the data of α and β diversity of gut microbial community, there were no significant differences among groups. Only some microbial compositions showed group differences. What are the differences in experimental design and findings between the present and previous study (Ding et al., 2021)? What levels of ammonia were used in the previous study? What age stage was the subject in the previous study? More discussion should be added for explaining possible reasons of different findings between studies.

 What’s the relationship between gut microbiota and liver metabolites? There are no differences in metabolism- or immunity-related bacterial genera in the present study. How did the author explain the changes in liver metabolites? More possible reasons may be provided in the discussion.

 How did ammonia exposure influence liver amino acid, nucleotide and lipid metabolism? The authors may give a simple implication about the possible mechanisms based on the data and references in the discussion.

Author Response

Li et al. evaluated the physiological, gut microbiomic and liver metabolomic responses to chronic exposure of different ammonia levels (0, 0.3, 3.0, and 20.0 mg/L) in juveniles of invasive turtle, Trachemys scripta elegans. They got the conclusion that ammonia exposure would significantly influence gut health, and disturb various metabolic pathways in aquatic turtle species. They measured gut microbiomic responses to chronic ammonia exposure, but did not test intestinal structure, so the conclusion should be specific.

Response 1: Revised. The sentence was changed to “……ammonia exposure would cause gut dysbacteriosis……”

This group has found that exposure to high concentrations of ammonia would cause marked toxic effects on various tissues including damaging intestinal structure and altering its microbial community in T. scripta elegans. The rationales to study responses to chronic ammonia exposure in juveniles of Trachemys scripta elegans should be mentioned in the introduction.

Response 2: Relevant statements were added at the end of the section of introduction.

In this study, the authors just tested the responses to ammonia levels in juveniles of invasive turtle and these results could not reflect the responses of all age stages, so juveniles should be present in the title.

Response 3: Revised. The word “juveniles” was added in the title.

The statistical analysis was missed in the part of Material and Methods. The authors should describe the statistical analysis as an independent part in details.

Response 4: Revised. The “data processing and analysis” was added in the part of Material and Methods.

What’s the statistical method used by the authors for analyzing the data of microbial composition? There are diverse bacteria in the gut and they can influence each other. Did they use adjusted P values?

Response 5: Non-parametric Kruskal-Wallis tests were used to test among-group differences in the relative abundance of gut microbial composition at each taxonomic level. Yes, we agreed that diverse gut bacteria could influence each other, although adjusted P values were not presented in the text.

It’s the similar condition for the metabolites. It’s better to correct the statistical analysis.

Response 6: One-way ANOVAs were used to test among-group differences in key identified metabolites (log10 transformed normalized intensity). We also agreed that different metabolites might influence each other, but we felt that it was no problem for performing one-way ANOVA on each metabolite because only a few metabolites were identified.

For the data of α and β diversity of gut microbial community, there were no significant differences among groups. Only some microbial compositions showed group differences. What are the differences in experimental design and findings between the present and previous study (Ding et al., 2021)? What levels of ammonia were used in the previous study? What age stage was the subject in the previous study? More discussion should be added for explaining possible reasons of different findings between studies.

Response 7: Revised. Yes, discrepant results between the present and previous study might be partly due to the differences in ammonia exposure level, age stage of experimental animals, rearing environmental condition, and so on (32 mg/L total ammonia-N and 152.33 ± 4.26 g body mass in the previous study).

What’s the relationship between gut microbiota and liver metabolites? There are no differences in metabolism- or immunity-related bacterial genera in the present study. How did the author explain the changes in liver metabolites? More possible reasons may be provided in the discussion.

Response 8: Tissue metabolites might be influenced by the changes in gut microbiota. But not all metabolites in the livers are influenced by gut bacteria, and the change in most liver metabolites should be due to liver metabolism itself. Actually, it could be predicted that the relationship between gut microbiota and gut metabolite would be more direct. So, we explain the change in liver metabolites mainly lies in ammonia-induced perturbations in metabolic pathways related to amino acid, nucleotide, lipid metabolism, or other metabolism in the livers. The relationship between gut microbiota and liver metabolites would be rather complicated, and was not necessarily simple linear. The correlation analysis was not directly done here to explore the potential link between gut bacterial community and liver metabolites, primarily due to relatively small changes in bacterial genera.

How did ammonia exposure influence liver amino acid, nucleotide and lipid metabolism? The authors may give a simple implication about the possible mechanisms based on the data and references in the discussion.

Response 9: Considering with the results reported in other species, ammonia exposure would alter the metabolism and secretion of ammonia in aquatic animals, and might disturb the biosynthesis and catabolism of amino acid, nucleotide, lipid, and so on. However, the relevant metabolic pathways involving amino acids nucleotide and lipid are rather complex, and were difficult to be clarified in this study. Additionally, discrepant findings would be displayed from different studies conducted on different aquatic species. For example, to increase the synthesis of glutamine would be expected under ammonia stress, and has been reported in some ammonia-exposed aquatic species. But a decrease in glutamine was found here, and that glutamine synthesis might also be hindered was used to explain in this study. Based on currently available data, we felt that to draw a common explanation might be still difficult.
